# DLUNet: Semi-supervised Learning based Dual-Light UNet for Multi-organ Segmentation

Haoran Lai[1], Tao Wang[2], and Shuoling Zhou[3]

[1] Guangdong Provincial Key Laboratory of Medical Image Processing, Southern Medical University, Guangzhou, 510515, China
hanranlai@163.com
[2] Guangdong Provincial Key Laboratory of Medical Image Processing, Southern Medical University, Guangzhou, 510515, China
wangtao_9802@sina.com
[3] Guangdong Provincial Key Laboratory of Medical Image Processing, Southern Medical University, Guangzhou, 510515, China
zslandsouling@163.com

**Abstract.** The manual ground truth of abdominal multi-organ is labor-intensive. In order to make full use of CT data, we developed a semi-supervised learning based dual-light UNet. In the training phase, it consists of two light UNets, which make full use of label and unlabeled data simultaneously by using consistent-based learning. Moreover, separable convolution and residual concatenation was introduced light UNet to reduce the computational cost. Further, a robust segmentation loss was applied to improve the performance. In the inference phase, only a light UNet is used, which required low time cost and less GPU memory utilization. The average DSC of this method in the validation set is 0.8718. The code is available in https://github.com/laihaoran/Semi-Supervised-nnUNet.

**Keywords:** Semi-supervised learning · UNet · Robust segmentation loss.

## 1 Introduction

Fast automatic abdominal multi-organs segmentation can greatly improve the labeling speed of radiologists. However, there are still a series of challenges for automatic abdominal multi-organ segmentation: 1) Manual labeling of ground truth requires significant labor cost. 2) There is a large amount of unlabeled data that can be used to improve performance. 3) Medical image segmentation suffers from unclear boundaries. 4) Integrated automatic segmentation algorithms need to meet the requirements of low time cost and less GPU memory utilization.

Semi-supervised learning can be achieved by combining a small amount of labeled data and a large amount of unlabeled data, thus enabling training on small labeled datasets. The current major semi-supervised learning algorithms can be categorized into 1) pseudo-labeling-based learning [1,6] and 2) consistency-based

learning [2,10]. The prospects of abdominal multi-organ segmentation have multiple categories and dense distribution (multiple categories may exist in a region), which is suitable for consistency-based learning.

Therefore, we propose a semi-supervised learning based dual-light UNet to achieve fast automatic abdominal multi-organs segmentation. First, consistency learning strategy was introduced in to the proposed network to effectively utilize the large amount of unlabeled data. Second, a light UNet was proposed to achieve efficient and fast automatic segmentation. Then, a robust segmentation loss function was applied to overcome the challenge of tiny foreground. Finally, this proposed method achieves fast and accurate automatic abdominal multi-organ segmentation.

The main contributions of this work are as follows.

- We use a network consistency-based semi-supervised learning strategy to leverage large amounts of unlabeled data.
- We propose a light UNet for fast and efficient automatic abdominal multi-organs segmentation.
- We adopt a robust segmentation loss function to effectively overcome the challenge of tiny foreground.

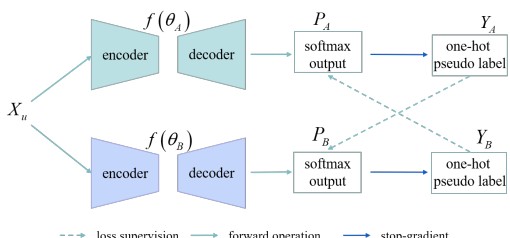

**Fig. 1.** Illustrating the architectures for consistent learning.

## 2 Method

### 2.1 Consistency-based learning

As shown in Figure 1, let the $X_l = \{x_{li}, i \in N\}$ and $X_u = \{x_{ui}, i \in M\}$ be the labeled and unlabeled data, respectively, where $N$ and $M$ are the number of labeled and unlabeled data, respectively. In our experiment, the condition of $\ll M$ is established for semi-supervised learning. First, dual identical networks $f(\theta_A)$ and $f(\theta_B)$ are built with different parameter initialization methods. Then, dual identical networks $f(\theta_A)$ and $f(\theta_B)$ are trained by using the labeled data for abdominal organ segmentation, respectively.

$$f(x_{li}; \theta_A) = p_{A,li}$$
$$f(x_{li}; \theta_B) = p_{B,li}$$
(1)

where $p$ is the probability map. Next, the trained network is used to obtain different probability map of unlabeled data and their pseudo-labels.

$$f(x_{ui}; \theta_A) = p_{A,ui}, f(x_{uj}; \theta_A) = p_{A,uj}$$
$$f(x_{ui}; \theta_B) = p_{B,ui}, f(x_{uj}; \theta_B) = p_{B,uj} \tag{2}$$

$$y_{A,ui} = \text{argmax}(p_{A,ui}), y_{A,uj} = \text{argmax}(p_{A,uj})$$
$$y_{B,ui} = \text{argmax}(p_{A,ui}), y_{B,uj} = \text{argmax}(p_{B,uj}) \tag{3}$$

CutMix operation [14] is implemented on different unlabeled data and pseudo labels:

$$x_{uij} = \mathbf{H} \odot x_{ui} + (1 - \mathbf{H}) \odot x_{uj}$$
$$y_{A,uij} = \mathbf{H} \odot y_{A,ui} + (1 - \mathbf{H}) \odot y_{A,uj} \tag{4}$$
$$y_{B,uij} = \mathbf{H} \odot y_{B,ui} + (1 - \mathbf{H}) \odot y_{B,uj}$$

In this situation, the outputs of the two networks can be used to supervise for each other, which achieves the network consistency-based learning.

$$f(x_{uij}; \theta_A) = p_{A,uij} \longrightarrow y_{B,uij}$$
$$f(x_{uij}; \theta_B) = p_{B,uij} \longrightarrow y_{A,uij} \tag{5}$$

During each iteration, the label data and the unlabel data are simultaneously input to the network for optimization.

## 2.2 Light UNet

To accelerate inference speed and reduce the GPU memory utilization, we modify the UNet in nnU-Net[9]. A light UNet was presented in Figure 2.

- We replace the original convolution with depthwise separable convolution [3], thus reducing the number of trainable parameters.
- Residual connection [5] was introduced between all convolution layers, including encoder and decoder, thus improving the representational ability of the UNet.

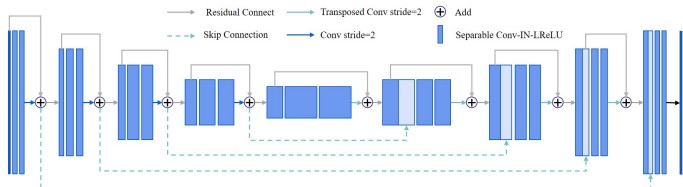

**Fig. 2.** The architecture of Light UNet.

### 2.3    Robust segmentation Loss

In the segmentation task, the commonly used segmentation loss is a combination of Dice loss and cross entropy (CE) loss, which have been proved be robust in various medical image segmentation task [11]. In this paper, based on the previous segmentation loss, the idea of mean absolute error (MAE) loss was introduced into Dice and CE loss respectively. Therefore, a robust segmentation loss fuction $\mathcal{L}_{RS}$ was proposed, which consists of noise robust dice loss $\mathcal{L}_{NRD}$ and taylor cross entropy loss $\mathcal{L}_{TCE}$.

$$\mathcal{L}_{RS} = \mathcal{L}_{NRD} + \mathcal{L}_{TCE} \tag{6}$$

$$\mathcal{L}_{NRD} = \frac{\sum_{n=1}^{DWH} |\mu_n - \upsilon_n|^{\gamma}}{\sum_{n=1}^{DWH} \mu_n^2 + \sum_{n=1}^{DWH} \upsilon_n^2 + \epsilon} \tag{7}$$

$$\mathcal{L}_{TCE} = \sum_{n=1}^{DWH} (1 - \mu_{n,\upsilon=1}) + \frac{\sum_{n=1}^{DWH} (1 - \mu_{n,\upsilon=1})^2}{2} \tag{8}$$

where $D$, $W$ and $H$ are the depth, width and height of input, respectively. $\mu$ and $\upsilon$ are the voxels of softmax output and ground truth, respectively.

### 2.4    Preprocessing and Inference

The dataset was preprocessed by nnU-Net configuration[9], including HU value clipping, HU values normalization, and resolution uniformity. In order to achieve category-balanced cropping for unlabeled data in training stage, a nnU-Net model was trained in advance using a small amount of labeled data. Then, a pseudo-label for unlabeled data is generated, which is only involved in achieving category-balanced cropping and not in other utilization.

In the inference phase, a patch shift-based approach was used to generate mask outputs for the entire 3D CT. We used 0.5 shift steps for each patch to alleviate the misclassification of the results by local information. Moreover, all patchs were flipped along three axes to generate robust performance.

## 3    Experiments

### 3.1    Dataset and evaluation measures

The FLARE2022 dataset is curated from more than 20 medical groups under the license permission, including MSD [13], KiTS [7,8], AbdomenCT-1K [12], and TCIA [4]. The training set includes 50 labelled CT scans with pancreas disease and 2000 unlabelled CT scans with liver, kidney, spleen, or pancreas diseases. The validation set includes 50 CT scans with liver, kidney, spleen, or pancreas diseases. The testing set includes 200 CT scans where 100 cases has liver, kidney, spleen, or pancreas diseases and the other 100 cases has uterine

corpus endometrial, urothelial bladder, stomach, sarcomas, or ovarian diseases. All the CT scans only have image information and the center information is not available.

The evaluation measures consist of two accuracy measures: Dice Similarity Coefficient (DSC) and Normalized Surface Dice (NSD), and three running efficiency measures: running time, area under GPU memory-time curve, and area under CPU utilization-time curve. Only DSC score was presented in the experiments. All measures will be used to compute the ranking. Moreover, the GPU memory consumption has a 2 GB tolerance.

### 3.2    Implementation details

**Environment settings** The development environments and requirements are presented in Table 1.

**Table 1.** Development environments and requirements.

| | |
|---|---|
| Windows/Ubuntu version | Ubuntu 18.04.5 LTS |
| CPU | Intel(R) Xeon(R) Gold 5218 CPU @ 2.30GHz |
| RAM | 503 GB |
| GPU (number and type) | Two NVIDIA RTX 2080Ti 11G |
| CUDA version | 11.0 |
| Programming language | Python 3.7 |
| Deep learning framework | Pytorch (Torch 1.11, torchvision 0.2.2) |

**Training protocols** Ther training protocols are presented in Table 2

**Table 2.** Training protocols.

| | |
|---|---|
| Network initialization | "he" normal initialization |
| Batch size | 1 |
| Patch size | 56×160×160 |
| Target resolution | 2.5×1.5×1.5 |
| Total epochs | 1000 |
| Optimizer | SGD with nesterov momentum ($\mu = 0.99$) |
| Initial learning rate (lr) | 0.01 |
| Lr decay schedule | halved by 200 epochs |
| Training time | 276 hours |
| Loss function | RRD + TCE |
| Number of model parameters | 5.59M |
| Number of flops | 33.81G |

## 4 Results and discussion

A public unlabeled validation set was used to evaluate the experiment results, which can be uploaded to the online[4] for metrics.

### 4.1 Ablation of semi-supervised learning

Table 3 shows the effects of introducing semi-supervised learning in the nnU-Net and light unet on the final segmentation performance, respectively. Two conclusions can be found from Table 3: (1) The segmentation performance of the light unet is inferior to the nnU-Net due to the less parameters, but the light unet can speed up the inference and reduce the GPU memory utilization. (2) The introduction of semi-supervised learning has greatly improved the segmentation performance for both. Further, the performance improvement is greater for the light unet with a smaller number of parameters than nnU-Net, which may be caused by model with few parameters has strong potential for improvement.

**Table 3.** Ablation of semi-supervised learning (SSL). LV, RK, SL, PC, AT, IVC, RAG, LAG, GB, EH, SM, DD, and LK are short for Liver, Right Kidney, Spleen, Pancreas, Aorta, Inferior Vena Cava, Right Adrenal Gland, Left Adrenal Gland, Gallbladder, Esophagus, Stomach, and Left kidney, respectively.

| Method | Mean | LV | RK | SL | PC | AT | IVC | RAG | LAG | GB | EH | SM | DD | LK |
|---|---|---|---|---|---|---|---|---|---|---|---|---|---|---|
| nnU-Net w/o SSL | 0.869 | 0.967 | 0.880 | 0.941 | 0.841 | 0.949 | 0.882 | 0.822 | 0.819 | 0.821 | 0.877 | 0.885 | 0.748 | 0.871 |
| nnU-Net w SSL | **0.895** | 0.978 | 0.897 | 0.973 | 0.909 | 0.973 | 0.922 | 0.839 | 0.826 | 0.779 | 0.900 | 0.914 | 0.838 | 0.888 |
| Light UNet w/o SSL | 0.837 | 0.965 | 0.869 | 0.932 | 0.830 | 0.945 | 0.860 | 0.766 | 0.731 | 0.731 | 0.837 | 0.858 | 0.717 | 0.843 |
| Light UNet w SSL | **0.878** | 0.976 | 0.910 | 0.969 | 0.894 | 0.960 | 0.896 | 0.807 | 0.763 | 0.764 | 0.865 | 0.915 | 0.799 | 0.891 |

**Table 4.** Comparison of loss function.

| Loss | Mean | LV | RK | SL | PC | AT | IVC | RAG | LAG | GB | EH | SM | DD | LK |
|---|---|---|---|---|---|---|---|---|---|---|---|---|---|---|
| Dice+CE | 0.869 | **0.972** | **0.915** | **0.954** | **0.861** | **0.958** | **0.884** | **0.823** | 0.814 | 0.720 | 0.867 | **0.888** | **0.751** | **0.889** |
| NRD+TCE | **0.870** | 0.967 | 0.880 | 0.941 | 0.841 | 0.949 | 0.882 | 0.822 | **0.819** | 0.821 | **0.877** | 0.885 | 0.748 | 0.871 |

### 4.2 Comparison of loss function

From Table Table 4, it can be found that the robust segmentation loss is superior to the combination of dice and CE loss in terms of overall performance. Moreover, it can be noticed that although the robust segmentation loss is inferior to the combination of dice and CE loss for the segmentation of most organs from

---

[4] https://flare22.grand-challenge.org/evaluation/challenge/submissions/create/

the segmentation performance of different organs, the robust segmentation loss has a great advantage for the segmentation of the gallbladder. The gallbladder belongs to the small target segmentation region, therefore, we conclude that robust segmentation loss has some advantages for the small target region.

### 4.3   Segmentation efficiency results

Considering the balance between segmentation performance and inference speed, we reduce the original 7 times flips in nnU-net to 3 tmes flips (tta). Moreover, in order to address the phenomenon that particularly large samples in the image will be out of memory during the inference process, we only keep the final generated labels and do not keep the intermediate network output (RAM). The result was performed in Table 5.

We did not upload docker to test computational efficiency issues. However, we tested on our own platform to test the optimization of computational efficiency. In the end, we achieved a test time of 0.67 hour on 50 validation samples, maximum ram is 18G, and GPU memory is 2045MB.

**Table 5.** Extra Processing for fianl result. IS(H) is short for inference speed, with hour as unit.

| Method | Mean | IS(H) | LV | RK | SL | PC | AT | IVC | RAG | LAG | GB | EH | SM | DD | LK |
|---|---|---|---|---|---|---|---|---|---|---|---|---|---|---|---|
| DLUNet | 0.878 | | 0.976 | 0.910 | 0.969 | 0.894 | 0.960 | 0.896 | 0.807 | 0.763 | 0.764 | 0.865 | 0.915 | 0.799 | 0.891 |
| DLUNet+tta | **0.884** | 2.00 | 0.977 | 0.910 | 0.972 | 0.899 | 0.962 | 0.901 | 0.816 | 0.762 | 0.801 | 0.873 | 0.917 | 0.800 | 0.895 |
| DLUNet+tta+RAM | 0.872 | **0.67** | 0.973 | 0.903 | 0.964 | 0.890 | 0.948 | 0.888 | 0.789 | 0.741 | 0.792 | 0.857 | 0.911 | 0.795 | 0.885 |



**Fig. 3.** Qualitative results on easy (case 06 and 21) and hard (case 47 and 48) examples. First column is the image, second column is the ground truth, third column is the predicted results by Light U-Net without ssl, third column is the predicted results by DLUNet with ssl.

### 4.4   Qualitative results

Figure 3 presents some easy and hard examples on validation set, and quantitative result is illustrated in Table 6. Comparing (Case 06 and Case 21) and (Case 47 and Case 48), we can find that our proposed method does not work well for lesion-affected organs. For example, the liver cancer region is wrongly identified in Case 47 and Case 48, especially Case 48. This situation may be due to our proposed method is implemented by a patch-based training strategy, which lacks global information.

**Table 6.** The DSC scores of easy and hard examples.

| Example | Method | Mean | LV | RK | SL | PC | AT | IVC | RAG | LAG | GB | EH | SM | DD | LK |
|---------|--------|------|----|----|----|----|----|-----|-----|-----|----|----|----|----|----|
| Case 06 | w/o ssl | 0.915 | 0.983 | 0.974 | 0.978 | 0.924 | 0.965 | 0.944 | 0.899 | 0.894 | 1.000 | 0.908 | 0.936 | 0.756 | 0.729 |
|         | w ssl   | **0.924** | 0.985 | 0.983 | 0.983 | 0.929 | 0.977 | 0.955 | 0.927 | 0.920 | 1.000 | 0.922 | 0.940 | 0.760 | 0.725 |
| Case 21 | w/o ssl | 0.946 | 0.985 | 0.972 | 0.983 | 0.926 | 0.966 | 0.937 | 0.869 | 0.864 | 1.000 | 0.935 | 0.969 | 0.926 | 0.973 |
|         | w ssl   | **0.957** | 0.988 | 0.980 | 0.989 | 0.932 | 0.980 | 0.946 | 0.894 | 0.897 | 1.000 | 0.949 | 0.973 | 0.936 | 0.981 |
| Case 47 | w/o ssl | 0.798 | 0.885 | 0.978 | 0.866 | 0.798 | 0.936 | **0.665** | **0.677** | 0.818 | **0.676** | 0.807 | 0.904 | **0.395** | 0.977 |
|         | w ssl   | **0.805** | 0.882 | 0.986 | 0.868 | 0.805 | 0.954 | **0.682** | 0.676 | 0.833 | **0.707** | 0.815 | 0.918 | **0.358** | 0.983 |
| Case 48 | w/o ssl | 0.716 | 0.971 | 0.971 | 0.667 | 0.841 | 0.958 | **0.461** | 0.679 | 0.856 | 0.000 | 0.693 | 0.598 | 0.796 | 0.811 |
|         | w ssl   | **0.729** | 0.972 | 0.978 | 0.702 | 0.861 | 0.970 | **0.456** | **0.747** | 0.869 | **0.000** | 0.692 | 0.623 | 0.795 | 0.812 |

### 4.5   The performance of testing set

As shown in Table 7, our method shows a competitive segmentation performance on the testing set. Moreover, we find that all metrics of case 97 are 0. This may be caused by the fact that the view of case 97 is flipped, which leads to the misjudgment of the inference optimization algorithm and terminates the inference in advance, resulting in not generating the correct segmentation output. Since the focus of our method is on segmentation performance improvement, the optimization of inference speed is neglected, resulting in the lack of advantage of our method in the final composite score.

**Table 7.** The performance of testing set.

| Metric | Mean | LV | RK | SL | PC | AT | IVC | RAG | LAG | GB | EH | SM | DD | LK |
|--------|------|----|----|----|----|----|-----|-----|-----|----|----|----|----|----|
| DSC | 0.881 | 0.968 | 0.941 | 0.949 | 0.854 | 0.949 | 0.900 | 0.815 | 0.805 | 0.809 | 0.805 | 0.924 | 0.797 | 0.937 |
| NSD | 0.940 | 0.969 | 0.960 | 0.961 | 0.954 | 0.982 | 0.923 | 0.953 | 0.939 | 0.828 | 0.913 | 0.951 | 0.926 | 0.958 |
| Times(s) | 73.92 | | | | | | | | | | | | | |
| AUC GPU | 138831 | | | | | | | | | | | | | |
| AUC CPU | 1195 | | | | | | | | | | | | | |

### 4.6   Limitation and future work

In this paper, we do not use existing deep learning model packaging techniques (e.g., TensorRT) to package the model, reduce computational memory, and in-

crease inference speed. Therefore, the implementation of the operation can be considered in the future work.

## 5   Conclusion

The FLARE2022 competition aims to design an efficient and accuracy abdominal multi-organ segmentation network by using a small amount of labeled data and a large amount of unlabeled data. In this paper, we proposed DLUNet for this task. First, consistent-based learning was introduced to achieve semi-supervised learning. Second, separable convolution and residual connection were used to greatly reduce the computational cost. Moreover, a robust segmentation loss was applied to improve segmentation performance. Experiments prove that the DLUNet achieves a certain balance in terms of model parameters, computation time, GPU memory utilization, and segmentation performance. The method is promising for the task.

**Acknowledgements** The authors of this paper declare that the segmentation method they implemented for participation in the FLARE 2022 challenge has not used any pre-trained models nor additional datasets other than those provided by the organizers. The proposed solution is fully automatic without any manual intervention.

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
