# OpenReview forum: "DLUNet: Semi-supervised Learning based Dual-Light UNet for Multi-organ Segmentation"
_MICCAI.org/2022/Challenge/FLARE_

### Official Review · Reviewer_LmvZ · 2022-09-17
**Efficient framework for semi-supervised work**

**Rating:** 8
**Confidence:** 4

**Review:**

Advantage:
1. A great robust work about semi-supervised learning design based dual-light UNet for labeled and unlabeled data.
2.  The inference is more efficient.

Disadvantage:
1. More details of the realization of semi-supervised learning process would be much more helpful for understanding the whole train and inference, respectively.

---

### Official Review · Reviewer_U3d4 · 2022-09-18
**Points are clear generally speaking, just need some minor changes**

**Rating:** 7
**Confidence:** 5

**Review:**

The paper's structure and grammar are good. It's very easy to read about. And I have some suggestion for you about the format.
1. The format of  Tabel 3.,Tabel 4. and the Tabel 5. makes me look inconvenient. and you can combine them together.
2. As for the Fig3. , you can add the case without ssl.

---

### Official Review · Reviewer_u4aQ · 2022-09-19
**a good article with enough innovations**

**Rating:** 9
**Confidence:** 5

**Review:**

Pros:
A dual-light UNet was designed and CutMix strategy was used for this semi-supervised learning task.
It was a good try for using consistent-based learning in terms of both the model and the data.
Modified loss function was tested.
nnUNet was modified to light Unet in order to have lower time cost.
Overall, it is a good article with enough innovations

Cons:
Advise to have more descriptions on preprocessing  phase and inference phase.

---

### Official Review · Reviewer_PWVJ · 2022-09-19
**Good paper, high scores**

**Rating:** 9
**Confidence:** 3

**Review:**

- Strong results
- Simple yet interesting ideas
- Concise description

Very good overall, although the tables could be more readable.

---

### Official Review · Reviewer_EdaL · 2022-09-20
**Overall good paper**

**Rating:** 8
**Confidence:** 4

**Review:**


1. The paper is well structured and easy to follow. The description of the method is clear.
2. The ablation study clearly shows how all parts of the methods affect segmentation quality and inference time.

---

### Public Comment · ~Zhengshan_Huang1 · 2022-09-21
**The article is clearly structured.**

The article is clearly structured but misses a description of the dataset.

---

### Meta-Review · Program_Chairs · 2022-09-28

**Recommendation:** Minor Revision
**Confidence:** 5

**Metareview:**

Nice paper. Please address the reviewers' comments in the revised manuscript.